# Observation and Analysis of Ejector Hysteresis Phenomena in the Hydrogen Recirculation Subsystem of PEMFCs

**DOI:** 10.3390/e25030426

**Published:** 2023-02-27

**Authors:** Mingyang Li, Mingxing Lin, Lei Wang, Yanbo Wang, Fengwen Pan, Xiaojun Zhao

**Affiliations:** 1National Demonstration Center for Experimental Mechanical Engineering Education (Shandong University), Key Laboratory of High-Efficiency and Clean Mechanical Manufacture of Ministry of Education, School of Mechanical Engineering, Shandong University, Jinan 250061, China; 2Shandong Guochuang Fuel Cell Technology Innovation Center Co., Ltd., Weifang 261061, China; 3School of Control Science and Engineering, Shandong University, Jinan 250061, China

**Keywords:** PEMFC, ejector, proportional valve, hysteresis, hydrogen recirculation

## Abstract

The optimization control and efficiency improvement of proton exchange membrane fuel cells (PEMFCs) are being paid more attention. Ejectors have been applied in PEMFC hydrogen recirculation subsystems due to the advantages of a simple structure and no power consumption. However, the hysteresis deviation of a proportional valve ejector is found in the loading and unloading processes such that the hysteresis phenomena can cause deviations in fuel cell control process and affect the power dynamic output stability of PEMFCs. This paper analyzes the causes and effects of proportional valve hysteresis phenomena through experiments and simulations. The results show that the resultant force of proportional valve armature is different in loading and unloading processes because of the hysteresis phenomena, and the maximum flow deviation is up to 0.42 g/s. The hysteresis phenomena of flow rate further cause a deviation of 68.7–89.3 kW in PEMFC power output. Finally, a control compensation model is proposed to effectively reduce the deviation. This study provides a reference for the control and optimization of PEMFC with ejector technology.

## 1. Introduction

The automotive industry consumes a large amount of fossil fuels, consequently exacerbating the global environmental pollution and energy crisis [1]. The proton exchange membrane fuel cell (PEMFC), as an energy conversion device that converts chemical energy in fuel and oxidant directly into electrical energy by an electrochemical reaction, has been widely used as the primary power source in transportation applications such as scooters, sedans, buses, boats and spacecraft. It has numerous advantages over the conventional power sources, such as a short startup time, compact system volume and relatively high system efficiency [2,3]. The fuel-cell electric vehicles (FCEVs) are promising alternatives meeting the requirements of clean energy and are helpful for solving the environmental and energy crises due to the advantages of high energy efficiency, no pollution and no carbon emission [4,5]. Therefore, the research on the control optimization and efficiency improvement of PEMFCs in the automotive field is popular.

As shown in Figure 1, a complete PEMFC system generally consists of a stack, a hydrogen supply system, an oxidant air supply system, a cooling system and other subsystems [6]. The humidified hydrogen and oxygen are fed into the fuel cell stack through the hydrogen and oxygen inlets, respectively [7]. The electrochemical reaction produces a large amount of heat, which is cooled by a coolant system. The thermal management subsystem maintains constant temperature of the stack by transmitting the generated heat to the outside of the stack and further into the environment [8,9]. The air subsystem is composed of a compressor, a humidifier and other components which compress and humidify air to the stack [10]. To be specific, the ejector-based anode hydrogen recirculation system is an effective method to improve the hydrogen utilization efficiency and to avoid the generated liquid water flooding in the anode channel [11]. The operation of the ejector recirculation device follows the power change process of the PEMFC system and changes the primary flow rate of the ejector by adjusting the proportional valve (PV). The output performance of the ejector varies with the loading and unloading requirements of the fuel cell stack. However, the loading and unloading curves of ejector often form a “hysteresis” during the actual operation process, which is one of the reasons for the difference or fluctuation of the output performance of fuel cell stacks under varying load conditions.

Frederik Stefanski [12] believed that the hysteresis phenomenon of proportional valves was affected by ferroelectric properties, which further affected the flow performance of proportional valves. Additionally, the authors compared the results with and without closed-loop compensation control in detail. The effect of “hysteresis” on dynamic flow performance of proportional valve can be eliminated by compensation control. For similar hysteresis phenomena and problems, Jiahai Huang [13] proposed a digital compensator in a hydraulic piezo-valve to reduce hysteresis and stabilize the dynamic output of the valve. Hysteresis phenomena of electromagnetic proportional valves have been studied in related literature, and the control compensation methods mainly focus on valve (parts) optimization. However, there are few studies on the influence of deviation and control compensation when the proportional valve is used in fuel-cell system.

In this paper, the hysteresis phenomenon in the fuel-cell system is systematically studied. A practical compensation scheme is proposed to eliminate the hydrogen supply deviation caused by the unilinear calibration of the proportional valve to improve the stability of the dynamic performance of the fuel cells’ operation.

## 2. Matching Selection and Experimental Test

Optimal design and advanced control of electrochemical reactants in PEMFC are essential to improving the durability and stability of commercial applications [14]. In general, excess supply of hydrogen and oxygen reactants to PEMFC’s electrochemical reaction flow field is necessary to prevent reactant starvation. Excess hydrogen is recirculated in a PEMFC’s flow field, which promotes gas mobility, and further minimizes the storage amount of liquid water generated by electrochemical reaction in the PEMFC’s flow field. In this way, it can avoid flooding and the degradation of the PEMFC’s performance. In addition, the higher reactant flow rate (recirculation amount) can keep the reactant concentration high and improve fuel utilization and fuel cell performance [15].

According to the calculation and matching, a fuel cell ejector which can fulfill the 100 kW level is selected by Equation (1). The theoretical fuel consumption of fuel cell is the hydrogen flow required by the electrochemical reaction of fuel-cell anode.
(1)$$m_{{react\_H}_{2}}=\frac{I_{stack}M_{H_{2}}}{2F}N_{cell}$$
where $m_{{react\_H}_{2}}$ is the mass flow of the reacted hydrogen;$\;I_{stack}$ is the stack current of the fuel cell; F is the Faraday constant, which is usually 96,500 C/mol;$\;M_{H_{2}}$ is the molar mass of hydrogen;${\;N}_{cell}$ is the number of cells in the stack, respectively.

The experimental process was as follows: the gas source supplies sufficient hydrogen and the inlet pressure was set to 12 bar; the outlet pressure was set to 1.5 bar. The rated control voltage of the proportional valve was 12 V, and the input current range of the proportional valve was from 0 to 1.8 A. The opening of the proportional valve was controlled by changing the input current of the proportional valve. During the loading and unloading process, the hydrogen mass flow rate corresponding to the input current value was collected. In the loading process, the minimum opening position current of the spool was about 1.08 A, and the maximum current of the spool was about 1.72 A. In the unloading process, the maximum position current of the spool was about 1.64 A, and the minimum opening position current was about 1.00 A.

The difference in flow rate controlled by a proportional valve in the loading and unloading processes was obtained by an experimental test. Figure 2 shows the difference in flow rates between loading and unloading calibration tests with the same control of input current. Table 1 shows the test result with a proportional valve for the ejector with primary flow pressure fixed at 12 bar and outlet pressure fixed at 1.5 bar.

## 3. Result Analysis of a Proportional Valve for the Ejector

The hydrogen cycling device used in this study was a combination of a proportional valve and an ejector to supply reactive hydrogen and recirculate unreactive hydrogen (Excess hydrogen) for the PEMFC anode subsystem, as shown in Figure 1.

The proportional valve was mainly composed of a spring, an armature, a coil and a flow section (gap). The displacement *x_i_* of the armature directly determines the gap area *S_i_* of hydrogen flowing through the proportional solenoid valve. The proportional valve flow rate for ejector was calculated by Equation (2).
(2)$$Q_{i}{=\mu }_{i}S_{i}\sqrt{\frac{2\cdot \Delta P_{i}}{\rho }}$$
where *Q_i_* is the flow rate; *S_i_* is the gap area; *μ_i_* is the flow coefficient; Δ*P* is the pressure drop at the control valve, which is calculated by the valve inlet pressure value minus the valve outlet pressure value; and *ρ* is fluid density.

As can be seen in the above, the gap area formed by armature’s movement is directly related to the flow rate, that is, the greater the distance armature moves upward, the greater the flow area. Therefore, it is particularly important to analyze the force and displacement characteristics of armature as follows.

When PEMFC is not running, the proportional valve is normally closed, and armature does not form flow gaps. Armature displacement changes with PEMFC running.

When the armature displacement increases, the force can be calculated by the balance force with Equation (3).

In this process, the electromagnetic force and fluid pressure have positive effects, promoting the armature movement. However, steady-state fluid power, armature gravity, spool friction, damping force, spring preload and spring force hinder the armature force have a negative effect.
(3)$$m\overset{˙˙}{x}=F_{m}+F_{p}-F_{w}-mg-f-c\dot{x}-F_{y}-kx$$

When the armature displacement decreases, the force can be calculated by the balance force with Equation (4).

In this process, the armature gravity, spring preload and spring force promote the armature movement in a positive way. On the contrary, the electromagnetic force, fluid pressure, steady fluid power and frictional force of the valve core and damping force become the forces hindering the armature movement, which is a negative effect.
(4)$$-m\overset{˙˙}{x}=F_{m}+F_{p}+Fw-mg+f+c\dot{x}-F_{y}-kx$$
where *m* is poppet mass, *F_m_* is electromagnetic force, *F_p_* is the hydraulic force, *F_w_* the steady-state hydraulic force, *f* is frictional resistance, *F_y_* is the preload of the spring, *c* is the damping coefficient, *k* is the stiffness of the spring and *x* is the displacement of the armature.

The following formula can be used to calculate the electromagnetic force (Equation (5)) theoretically:(5)$$F_{m}{=k}_{m}i-k_{y}x$$
where *k_m_* is the input current gain of the solenoid proportional valve; *i* is the input current; *k_y_* is the stiffness of the spring at the *x* position.

According to the analysis, when the proportional valve armature moves in the linear zone, the output force has the following characteristics:(1)Electromagnetic hysteresis: The electromagnetic force is affected by the hysteresis of the magnetization properties of magnetic materials. Specifically, when the control current changes in reciprocating fashion, there are some differences in the electromagnetic suction corresponding to the same current.(2)Friction hysteresis: Mainly due to the influence of eccentricity and friction coefficient of armature and guide sleeve, the existence of light eccentric force will also produce radial clamping force, which increases the friction hysteresis.(3)Time lag or delay effect: the main dynamic suction changes lag current changes; the situation is generally a rare or weak influence.

The results of the above analysis are shown in Figure 3. The difference in the resultant force of armature in the process of loading and unloading caused hysteresis, which resulted in different flow gaps and ultimately affected the flow rate.

## 4. Simulation and Analysis of Influence on PEMFC Calibration

### 4.1. Introduction of Simulation Model

Using the CRUISE_M software of AVL company, the model of the anode subsystem and cathode subsystem was established by accessing the software case library and tool library.

### 4.2. Anode Subsystem Model

As shown in the Figure 4, the anode subsystem model mainly included a hydrogen source, an ejector, a manifold, a water separator, a purge valve and other modules. The unreacted hydrogen in the anode system was recirculated with the ejector. The hydrogen supply proportional valve model used the boundary module directly and used the test data as the input for the primary flow characteristics of the ejector. The water separator module separated the liquid water from the anode flow field, and the purge valve module purged impurities and other gaseous substances.

### 4.3. Cathode Subsystem Model

The cathode subsystem is mainly composed of an air compressor module, a humidifier module and a manifold module.

### 4.4. Main Theoretical Calculation and Results

The theoretical fuel consumption of fuel cell is the hydrogen flow required by the electrochemical reaction of fuel cell anode. The stack current is closely related to the hydrogen supply mass flow, so the hydrogen supply also determines the stack output current. The target output current in the simulation is calculated by Equation (6).
(6)$$I_{stack}=2\;F\frac{m_{{react\_H}_{2}}}{M_{H_{2}}N_{cell}}$$
where $m_{{react\_H}_{2}}$ is the mass flow of the reacted hydrogen; $I_{stack}$ is the current of the fuel cell stack; F is the Faraday constant, which is usually 96,500 C/mol; $M_{H_{2}}$ is the molar mass of hydrogen; $N_{cell}$ is the number of cells in the stack.

The voltage of the stack is calculated by Equation (7).
(7)$$U_{stack}=E_{cell}N_{cell}$$
where $U_{stack}$ is the voltage of the stack; $E_{cell}$ is the voltage of cell or desired electric potential; $N_{cell}$ is the number of cells in the stack.

The power of the stack is defined as Equation (8).
(8)$$P_{stack}=P_{cell}N_{cell}=U_{cell}I_{stack}N_{cell}$$
where $P_{stack}$ is the power of the stack; $P_{cell}$ is the power of single fuel cell;$\;U_{cell}$ is the voltage of the single fuel cell.

The electrochemical reaction pressure and stoichiometric ratio of fuel cells cause a difference in fuel-cell performance. In the process of fuel-cell control calibration, if the same opening is taken as the target control, the hysteresis is different in the processes of loading and unloading. The hydrogen supply and output performance of fuel cells should be emphatically analyzed. The basic reaction electromotive force is shown in Equation (9). This voltage parameter is usually used as an electrochemical reaction ideal potential or open-circuit potential.
(9)$$E_{cell}{=E}_{0}+\frac{RT}{N_{cell}F}ln(\frac{P_{H_{2}}P_{O_{2}}^{0.5}}{P_{H_{2O}}})$$
where $E_{cell}$ is the voltage of cell or desired electric potential; $E_{0}$ is the theoretical electric potential; R is the constant of gas, which is usually 8.314 J/mol/K; T is the temperature in Kelvin; $N_{cell}$ is the number of electron transfers; F is Faraday’s constant, which is usually 96,500 C/mol; $P_{H_{2}}$ is the partial pressure of hydrogen; $P_{O_{2}}$ is the partial pressure of oxygen; $P_{H_{2O}}$ is the gaseous moisture pressure.

There is a characteristic relationship between stack voltage and current in fuel cells. When stack current increases, the voltage decreases, which is mainly due to the voltage loss caused by electrochemical activation, concentration polarization and internal resistance. The voltage in the simulated dynamic change process was calculated by Equation (10).
(10)$$U_{cell}=E_{cell}-u_{0}-J_{0}r_{\Omega }=E_{cell}-\mu_{0}-\frac{I_{stack}}{A_{area}}r_{\Omega }$$
where $U_{cell}$ is the voltage of the single fuel cell; $u_{0}$ is the cathode voltage loss; $J_{0}$ is the stack current density; $r_{\Omega }$ is the ohmic resistance of the fuel cell; $A_{area}$ is the fuel-cell area.

The basic flow chart of simulation analysis of the impact on fuel cells is shown in Figure 5.

The proportional valve controls the primary flow of the ejector. According to the law of conservation of matter, the proportional valve flow is equal to the primary flow. Proportional valves also control the ejector outlet and fuel-cell-stack inlet pressure.

It can be seen in Table 1 that when the control current of the proportional valve is 1.32 A, the hydrogen flow rate is 0.97 g/s in loading and 1.37 g/s in unloading. In Figure 6, simulation results show that the ejector flow changes with the changes in control. The ejector hydrogen outlet is connected to the stack inlet, and the stability of stack pressure can be ensured by keeping the ejector outlet pressure stable. In the model, the pressure can be stabilized by adjusting the PID of the pressure regulating valve. The pressure at the ejector outlet and the stack in the simulation model can basically stabilize at the demand pressure of 1.5 bar.

As shown in Figure 7, The same proportional valve inputs current to control flow and pressure when the pressure is constant and the loading and unloading flows vary. 

The above deviation often occurs when the control current of proportional valve is calibrated by a single curve. As shown in Figure 8, a bilinear lookup curve can be used to compensate dynamic deviation. According to the change in stack current, the stack is in a loading stage or unloading stage. If the stack current does not change, the current PV control current is kept. If the stack current changes, the control current output of loading or unloading is switched. In Figure 9, when the input current of the proportional valve 1.32 A, the output power of the fuel cell stack is 68.7 kW in loading and 89.3 kW in unloading, and output power deviation of the fuel cell stack is obvious.

As shown in Figure 9, without compensation control, when the PV control current is 1.32 A, the deviation between power point ① and power point ② is obvious. If combined with the compensation method in Figure 8, the compensation PV control current becomes 1.23 A in unloading, and the hydrogen supply flow rate at power point ③ is about 0.97 g/s. The power point ③ after compensation is the same as that of power point ①, and the compensation mode can ensure the stability of stack-power output.

## 5. Conclusions

In this paper, the ejector hydrogen circulation device with a proportional valve was matched for 100 kW PEMFC. In the process of testing and calibration, the differential flow rate of the proportional valve in loading and unloading processes was analyzed. The influence of hysteresis caused by loading and unloading on the PEMFC was studied. The main results were as follows:(1)The ejector with a proportional valve can supply hydrogen and recycle hydrogen, which can replace the traditional active hydrogen circulating pump.(2)The difference in flow rate of the ejector from a proportional valve in a calibration test was verified in the experiment. The hysteresis caused by different flow rates leads to the deviation of the primary flow of the ejector. Additionally, the main reason for hysteresis is the force of proportional valve armature in the loading and unloading processes.(3)The simulation results show that the hysteresis causes the deviation of PEMFCs’ dynamic performance. If there is no control compensation measure, the stack output power is higher in unloading process than in loading process. Therefore, it is necessary to reduce the dynamic deviation of ejector primary flow by controlling compensation.(4)The PV control current can be calculated by a double lookup curve in PEMFC system calibration, which can effectively reduce the primary flow deviation of ejector. The content of this paper can provide reference for the subsequent development and research of PEMFCs.

## Figures and Tables

**Figure 1 entropy-25-00426-f001:**
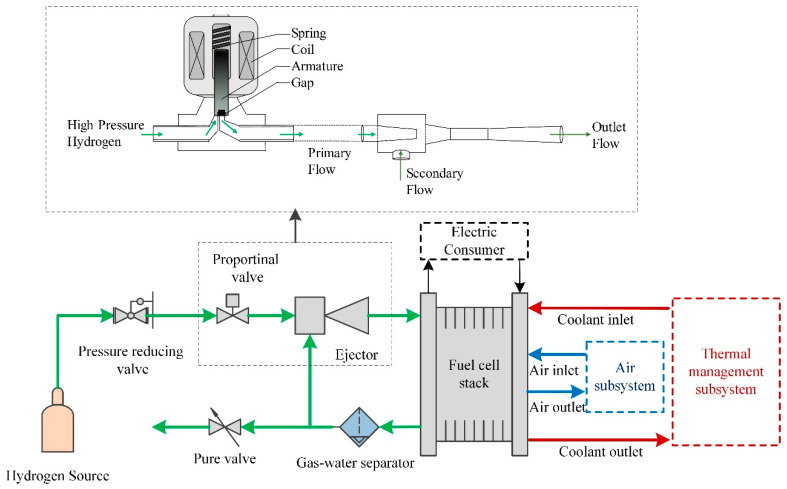
PEMFC system and hydrogen subsystem with proportional valve and ejector.

**Figure 2 entropy-25-00426-f002:**
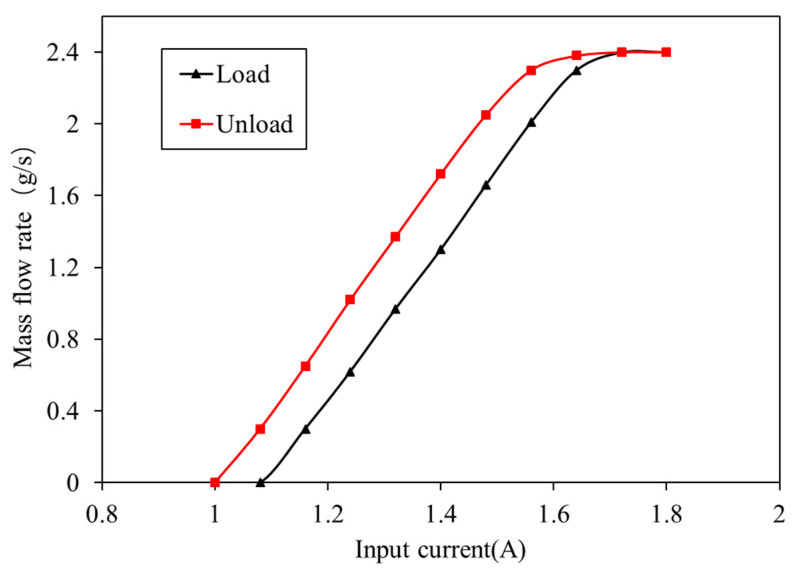
Flow rate of ejector with a proportional valve obtained by experiment.

**Figure 3 entropy-25-00426-f003:**
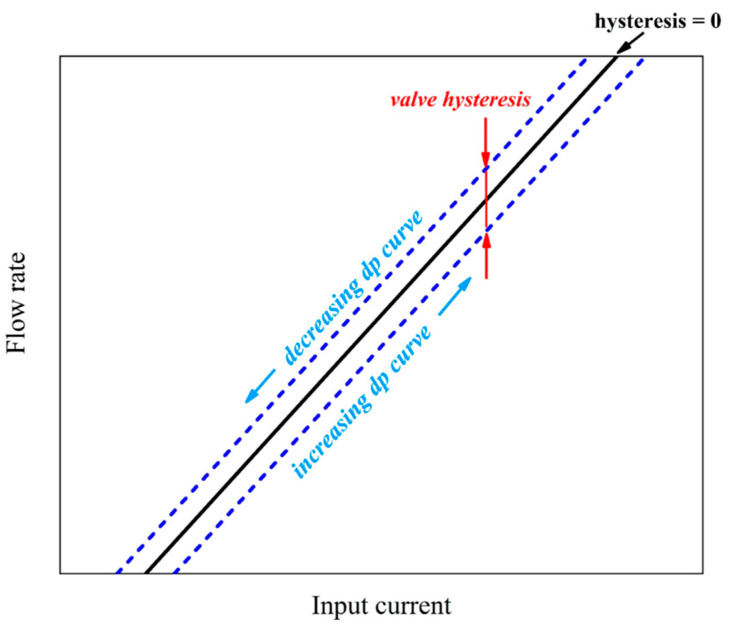
Schematic diagram of flow hysteresis when a proportional valve keeps the same input controls and current, and different forces.

**Figure 4 entropy-25-00426-f004:**
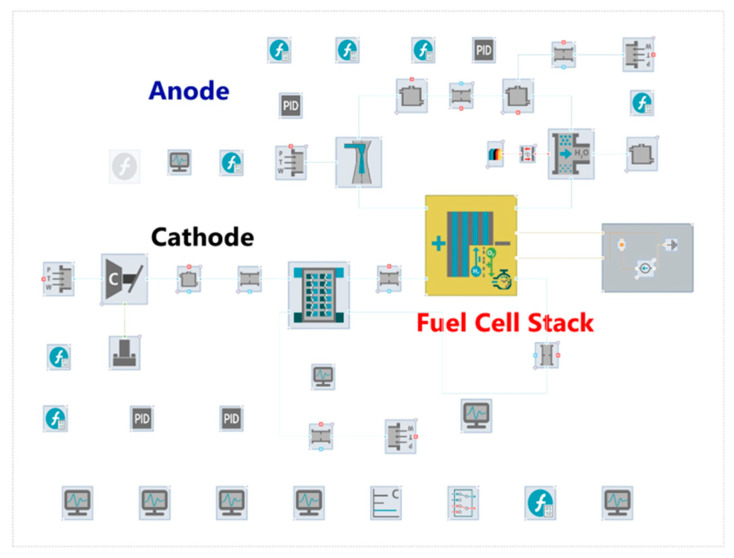
Simulation model of the cathode and anode subsystem of the fuel cell.

**Figure 5 entropy-25-00426-f005:**
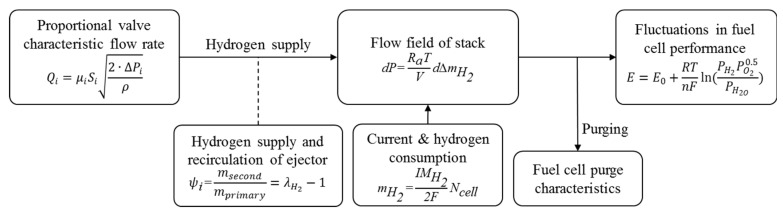
Flow chart of impact analysis on PEMFC.

**Figure 6 entropy-25-00426-f006:**
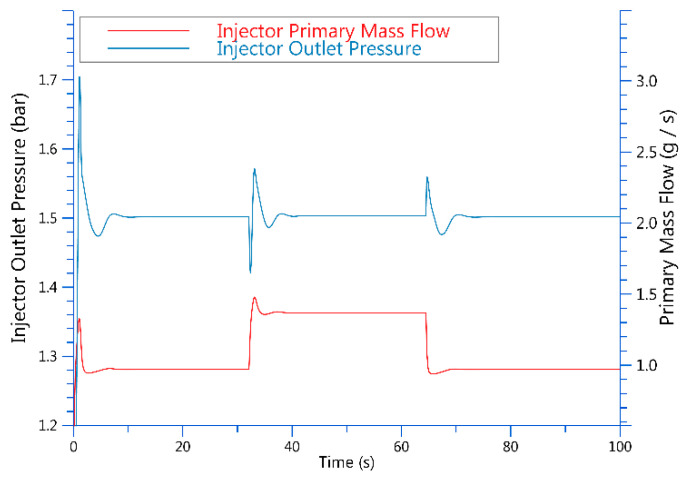
Simulation results of primary-flow and ejector-outlet pressure control.

**Figure 7 entropy-25-00426-f007:**
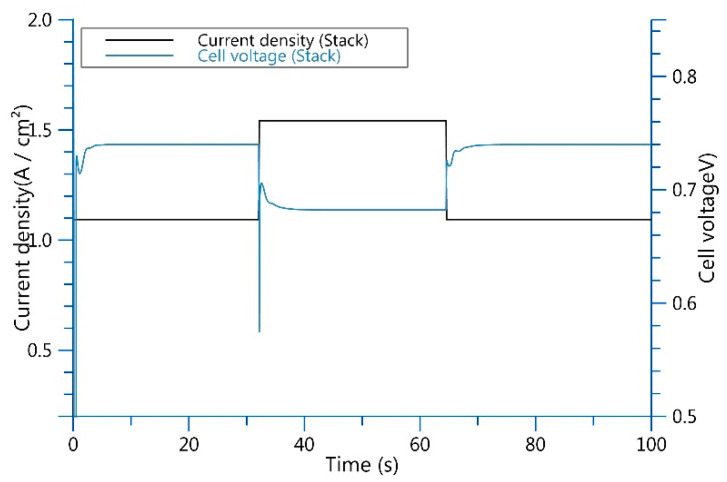
A different hydrogen flow rate causes different stack-output current and voltage.

**Figure 8 entropy-25-00426-f008:**
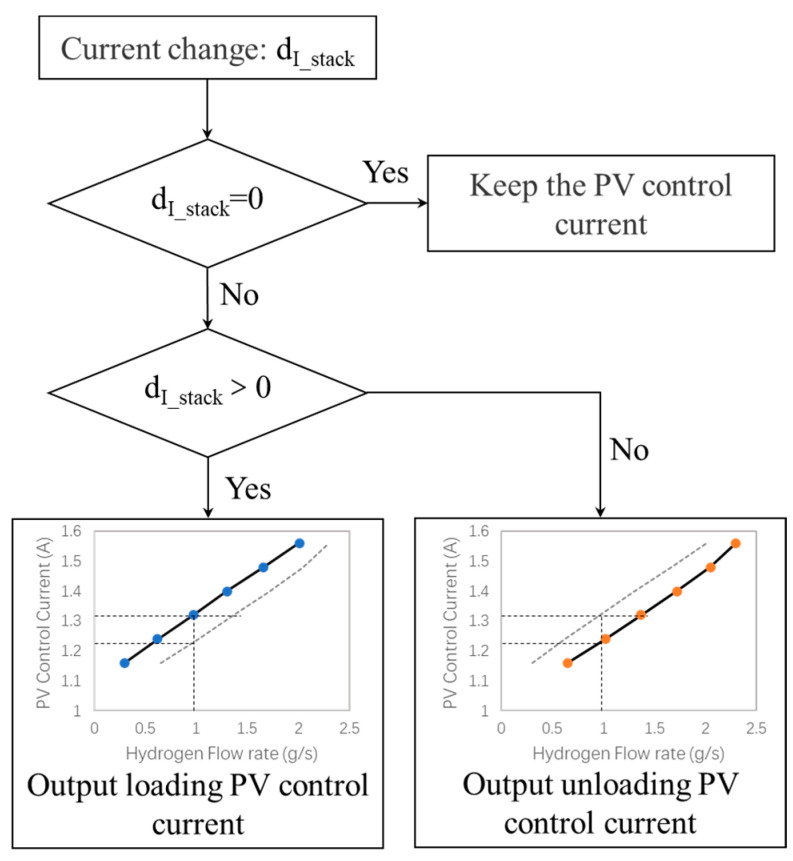
One of the proportional valve dynamic control compensation modes.

**Figure 9 entropy-25-00426-f009:**
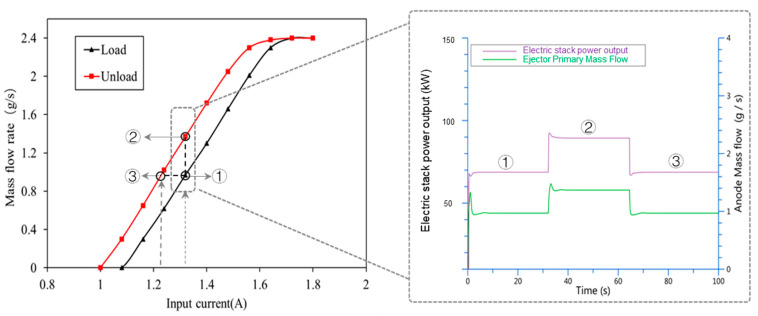
Influence of flow deviation on performance results of PEMFC simulation.

**Table 1 entropy-25-00426-t001:** Test data with a proportional valve for the ejector.

Input Current (A)	Mass Flow Rate of Fuel Cell Load (g/s)	Mass Flow Rate of Fuel Cell Unload (g/s)
1	0.01	0.00
1.08	0.01	0.30
1.16	0.30	0.65
1.24	0.62	1.02
1.32	0.97	1.37
1.4	1.30	1.72
1.48	1.66	2.05
1.56	2.01	2.30
1.64	2.30	2.38
1.72	2.40	2.40
1.8	2.40	2.40

## Data Availability

No new data were created or analyzed in this study. Data sharing is not applicable to this article.

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
