# Peer review of "Observation and Analysis of Ejector Hysteresis Phenomena in the Hydrogen Recirculation Subsystem of PEMFCs"

_entropy, 2023, doi:10.3390/e25030426_

Round 1

Reviewer 1 Report

This work by Mingyang Li et al. lacks enough explanation of the results and is filled with well-known equations in the field and explanations of the variables in the equations. The achievements provided in the conclusion are not thoroughly and in-depth explained in the paper. The only reasonable part of the paper is its introduction, which can be improved to highlight the novelty of the authors’ work. This paper may not be approved by Entropy unless clear and thorough results and discussion are provided.

Here are a few specific examples of the issues in writing. However, the overview of this paper is not acceptable.

1.     Line 112-114, please improve the writing to make sentences more concise and connected.

2.     Line 210: one equation is repeatedly provided (equations 1 and 6)

3.     Page 7 is basically an explanation of the well-known equations without clear information on how the model contributes to the cell voltage. The only information regarding the model of study is provided in Figure 5 with no specific description.

4.     The main body of the paper contains just three plots of results and non is explained in depth.

Reviewer 2 Report

The present manuscript reports the combined modelling-experiment investigation of the behaviour of the ejector component of a fuel cell system, The authors investigated the difference and hysteresis behaviour introduced during the loading and unloading stages. The study demonstrates that the resultant force of proportional valve armature is different during loading and unloading, which leads to different valve gaps and flow deviations and ultimately to subpar performance of the flow cell. Based on the provided data and experimental observations the work concludes that this valve behaviour and consequently the overall performance of the fuel cell can be controlled by real-time dynamic regulation to compensate for the observed differences during loading and unloading. The manuscript is in general well written and referenced. I am happy to recommend the publication of the present manuscript after minor revision.

A very minor point for the authors to correct:

Page 3, line 85; “…which further affected the flow performance. And detailed
comparison of closed loop control with and without hysteresis compensation.” Please revise.

Reviewer 3 Report

This manuscript analyzed the ejector hysteresis phenomena in the PEMFC hydrogen subsystem. There are several major problems to be addressed:

1. The abstract does not need to discuss too much background, but should pay more attention to the research method and results of this paper.

2. The last paragraph of the introduction should clearly explain the research purpose, novelty and contribution.

3. The model in this manuscript lacks reliable verification.

4. The research results presented are too few, and only simple results. There is not much meaningful analysis, let alone measures to improve its hysteresis. 

Round 2

Reviewer 3 Report

The authors could not adequately respond to my comment 3. The study is not bad enough to be rejected, it can be rejected or accepted by taking into account the other referee comments of the esteemed editor.